# Effect of Herbaceous Layer Interference on the Post-Fire Regeneration of a Serotinous Pine (*Pinus pinaster* Aiton) across Two Seedling Ages

**Jorge Castro [1,*]** and **Alexandro B. Leverkus [2]**

[1] Departamento de Ecología, Facultad de Ciencias, Universidad de Granada, Av. Fuentenueva SN, 18071 Granada, Spain

[2] Field Station Fabrikschleichach, Department of Animal Ecology and Tropical Biology (Zoology III), Julius-Maximilians-University Würzburg, 96181 Rauhenebrach, Germany; leverkus@ugr.es

[*] Correspondence: jorge@ugr.es; Tel.: +34-958-241-000 (ext. 20098)

**Abstract:** Herbaceous vegetation is a major source of interference with the regeneration of woody species. This is particularly the case after forest fires, as a dense herbaceous layer usually regenerates naturally. Although the competitive effect of the herbaceous vegetation upon tree seedlings has been widely studied, there are still gaps in knowledge for management related to the effect of tree seedling age and size on the outcome of the interaction. In this study, we seek to determine the response of maritime pine (*Pinus pinaster* Aiton) seedlings to herbaceous competition at two different seedling ages. For that, two treatments of herbaceous competition were implemented, namely unweeded (no action around pine seedlings) and weeded (herbaceous cover removed around pine seedlings). Treatments were conducted twice (2 and 4 years after the fire), and we monitored seedling survival and growth at the end of each growing season. The treatments were implemented across three adjacent landscape units that differed in the management of burned wood and that are representative of common post-fire scenarios: no intervention, salvage logging, and an intermediate degree of intervention. Weeding increased seedling survival from 44.7% to 67.8% when seedlings were 2 years old, but had no effect for four-year-old seedlings, which showed 99% survival. Seedling growth also increased in the weeding treatment, but only slightly. Moreover, growth (and survival for two-year-old seedlings) was strongly correlated with initial seedling size, particularly in the case of two-year-old seedlings. Initial pine seedling height was strongly and positively correlated with the height of the herbaceous layer, supporting the existence of microsite features that promote plant growth above competitive effects. The results support that management actions conducive to foster post-fire pine forest restoration in this Mediterranean ecosystem should reduce herbaceous competition at early stages after fire (second or third year) and focus on larger seedlings.

**Keywords:** burnt wood management; wildfire management; Mediterranean-type ecosystems; plant demography; plant competition; post-fire regeneration; salvage logging

## 1. Introduction

Competition with herbaceous vegetation is a major constraint for woody species recruitment, both through natural regeneration [1–3] and in reforestations and afforestations [4–7]. Competition with herbs can be particularly limiting for the growth and survival of woody seedlings after wildfire, which usually removes competitive tree canopies but triggers the establishment of dense herbaceous cover (e.g., [8–10]). As a consequence, the control of the herbaceous cover is an old topic of concern in forestry, particularly after fire, and large efforts are devoted to optimize its management and override its negative impact [1,4,11].

The mechanisms underlying the negative effect of the herbaceous cover on tree seedling performance are mainly mediated through competition for light, water, nutrients, or a combination of these factors [5,12–14], although other mechanisms may also be involved (e.g., allelopathy; [15]). The negative effect of herbaceous cover is quite ubiquitous, and although its effect may change depending on environmental conditions (even turning to facilitation in some cases; [16–18]), a competitive interaction has been widely reported for biomes and ecosystem types as different as conifer forests [2], broadleaf forests [5,19], grasslands [6,13,20], and Mediterranean-type ecosystems [21,22]. However, despite clear patterns in this interaction, there are still gaps of knowledge that are relevant for post-fire forest restoration. In particular, the time span in which the interference with the herbaceous layer may constrain tree regeneration is critical to plan appropriate management actions. Given that ground vegetation and tree seedlings compete for resources, any niche separation of resource use, either above or below ground, may reduce the negative effect of the competition by herbs. Thus, the negative effect of the herbaceous layer should be expected to decrease with increasing tree size, as both the roots and the canopy usually grow deeper and taller, respectively, than those of herbs. In fact, competition by herbs tends to be greater at the early establishment phases (when niche overlap is maximum), whereas at sapling or adult stage the direction of the interaction may shift against the herbaceous layer (negative effect of trees on understory vegetation; [23,24]).

Anthropogenic management after forest disturbances may further affect the development of the herbaceous layer through modifications of light, nutrient availability, soil moisture, and other microsite attributes (e.g., [25,26]), and thus modulate the interference exerted by herbs on tree seedlings. Post-fire management is particularly a case where human decisions may create habitat types that contrast in overall environmental conditions. After forest fires, restoration activities require a decision on how to manage the fire-damaged trees, which generally involves two contrasting options: removing burned trees through post-fire salvage logging or leaving them standing following non-intervention policies [26,27]. These two options, as well as any intermediate degree of intervention, create habitat types differing in physical structure but also in light, nutrient, and soil moisture conditions [25,28–30], which may affect the development of both the herbaceous layer and tree seedlings. In fact, the rapid regeneration of a herbaceous layer is a common pattern after fires in many forest types (e.g., [9,31]). It is thus relevant to understand how this layer may affect the regeneration of trees across stand-scale post-fire management actions, and how the timing of interventions aimed at reducing competition on seedlings may affect the interaction between the herbs and the regenerating trees.

In this study, we monitored the effect of the herbaceous layer on maritime pine (*Pinus pinaster* Aiton) seedling performance at two different-aged seedlings in three landscape units that received different post-fire management. The study was conducted in a burned area in Southern Spain where maritime pine regenerated naturally. Two years after the fire, the area had a high herbaceous cover, and pine seedlings (2 years old at that time) were subjected to two levels of weed removal (unweeded and weeded). After two additional years (four-year-old saplings), the weeding treatments were repeated on different seedlings. We hypothesized that (i) the herbaceous layer would have a deleterious effect on pine regeneration both in terms of survival and performance, (ii) the negative effect of the herbaceous layer would decline with greater seedling age, and (iii) larger seedlings, regardless of their age, would be less sensitive to competition from herbs. Overall, this study should help elucidate the effects of competition on tree regeneration in the context of different post-fire forest management treatments. Ultimately, we aim to provide managers with recommendations on the effectiveness of interventions that seek to reduce the effects of herb competition on tree regeneration, including any effects of timing and initial seedling size, and with applicability to three different management scenarios after fires.

## 2. Material and Methods

### 2.1. Study Site and Species

The study site is located in the Sierra Nevada Natural Park (southeast Spain), where a fire burned 1300 ha of pine stands in September 2005 [28]. The site was part of a *P. pinaster* reforestation of approximately 40 ha, located at around 1400 m a.s.l. (36°57′9″ N, 3°29′36″ W). Tree density (measured after the fire) was 1477 ± 46 individuals per ha, with a basal trunk diameter of 17.7 ± 0.2 cm (mean ± SE; Table 1). The site is located on an southwest-oriented hillside (average slope: 30.3 ± 5.7%) with micaschist as bedrock. Climate in the area is Mediterranean, with warm, dry summers and mild, rainy winters. Mean annual precipitation at the site is 487 ± 50 mm (1988–2011) and mean annual temperature is 11.8 ± 0.5 °C at 1652 m a.s.l. (period 1994–2011; State Meteorological Agency, Spanish Ministry of Environment).

**Table 1.** Characteristics of the landscape units used in this study and initial values of herb and pine seedlings growth parameters for each of these units.

| | | Landscape Unit | | |
| --- | --- | --- | --- | --- |
| | | NI | PCL | SL |
| Sampling unit variables | Centroid coordinates ($x$, $y$) | 456,142 4,090,020 | 455,887 4,089,880 | 456,024 4,089,940 |
| | Area (m$^2$) | 18,798 | 14,586 | 26,157 |
| | Elevation (m a.s.l.) | 1533 | 1430 | 1474 |
| | Slope (%) [a] | 26.8 | 23.9 | 23.1 |
| | Pre-treatment tree density (individuals/ha) [b] | 1304 ± 95 | 1236 ± 73 | 1316 ± 89 |
| | Pre-treatment tree height (m) [c] | 5.8 ± 0.2 | 6.1 ± 0.2 | n.a. |
| | Pre-treatment tree basal diameter (cm) [d] | 18.9 ± 0.6 | 20.1 ± 0.4 | 18.8 ± 0.7 |
| Initial values of herbaceous cover and pine seedlings | Herb height year 2 (cm) | 15.7 ± 1.0 | 15.3±0.9 | 11.8 ± 1.0 |
| | Pine height year 2 (cm) | 17.6 ± 0.6 | 24.1±0.7 | 18.2 ± 0.6 |
| | Pine stem diameter year 2 (mm) | 4.3 ± 0.2 | 6.5 ± 0.2 | 5.9 ± 0.2 |
| | Herb height year 4 (cm) | 24.6 ± 1.6 | 25.8 ± 1.6 | 16.7 ± 1.6 |
| | Pine height year 4 (cm) | 50.5 ± 1.2 | 47.5 ± 1.2 | 40.5 ± 1.2 |
| | Pine stem diameter year 4 (mm) | 11.4 ± 0.3 | 12.1 ± 0.4 | 12.4 ± 0.4 |

[a] Slope was calculated as a single value for the whole replicate from a digital elevation model (data provided by the local Forest Service). [b] Pre-treatment tree density was estimated by counting the trees in four 25 × 25 m randomly placed quadrats per landscape unit. [c] Pre-treatment tree height was estimated from 30 randomly trees per landscape unit. [d] Pre-treatment basal tree diameter was estimated for 120 random trees per landscape unit. n.a. = not available. NI = non-intervention unit; PCL = partial cut plus lopping unit; SL = salvage logging unit.

*P. pinaster* grows in the Western Mediterranean Basin and the Atlantic area of the Iberian Peninsula and Southern France, from sea level to about 1700 m a.s.l. It is a fast-growing species that has been widely used in reforestation planting, thereby increasing its distribution area in the Mediterranean Basin throughout the 20th century. It produces serotinous cones that protect the seeds from intense heat [32]. Seeds may still be viable after short heat pulses of above 100 °C [33], and the regeneration of the species after fire relies mostly on the aerial seed bank. Abundant *P. pinaster* seedling regeneration occurred naturally in the study area after the fire, with seedling emergence occurring mostly in late February 2006 (about 6 months after the fire; [28]).

The whole pine stand results from a reforestation conducted around 50 years before the fire on terraces made with bulldozers—previously a common reforestation practice on hillsides in Spain. Each terrace stairstep is composed of a steep backslope (approximately 90 cm high), and a nearly flat area approximately 3 m wide. Initial seedling density was very low on backslopes [28], so the study of the effect of herbaceous competition was conducted with seedlings located on the flat terrace areas.

## 2.2. Experimental Design

Several post-fire management regimes were experimentally implemented, thereby generating distinctive landscape units across the burned area. This study was conducted in three landscape units, each generated after one of the following management interventions:

1.  "Non-intervention" (NI), where no post-fire intervention was conducted and all burned trees were left standing.
2.  "Partial cut plus lopping" (PCL), generated by the felling of about 90% of the trees, with the main branches also lopped off, but leaving all the cut biomass (boles and branches) in situ on the ground. After treatment application, felled logs and branches covered 45% of the surface at 0–10 cm from the ground, 61% at 11–50 cm, and 9% at 51–100 cm [28].
3.  "Salvage logging" (SL), where trees were felled and limbed with the use of chainsaws. Woody debris was masticated using a tractor, and trunks were manually piled in groups of 10–15. The local Forest Service planned to remove the piled trunks with a log forwarder in this SL treatment, but this step was later cancelled due to difficulties in precisely operating machinery within the spatial arrangement of the plots.

The three management treatments were implemented by the local Forest Service from 21 April 2006 to 10 May 2006 (ca. seven months after the fire) in one of three adjacent plots of ca. 2 ha ([28]; Figure 1; Table 1). They form part of a long-term study on the effect of post-fire management on forest regeneration, and they differ in physical structure related to the amount and characteristics of the burned woody debris as a result of management ([28]; Table 1). Whereas the full experiment includes several replicate plots of each management treatment [9], only one replicate of each treatment was initially dominated by *P. pinaster* and harbored sufficient density of pine seedlings to study their regeneration after the fire [28]. As a result, we do not aim to make statistical inferences here on the effect of post-fire management treatments (as that would constitute pseudoreplication); rather, we aim to test the effect of managing competition by herbs throughout these three landscape units and thereby broaden the applicability of our results to a set of landscape conditions [34,35]. In any case, the three landscape units used in this study were similarly sized within an otherwise homogeneous landscape setting, presenting equal pre-treatment conditions in terms of bedrock (micaschist), slope, aspect (southwest exposure), high fire severity, and similar stand characteristics ([28]; see Table 1 for further details). To reduce spatial autocorrelation, we monitored a large number of seedlings (600 in total, see below) that were scattered throughout the surface of the three landscape units.

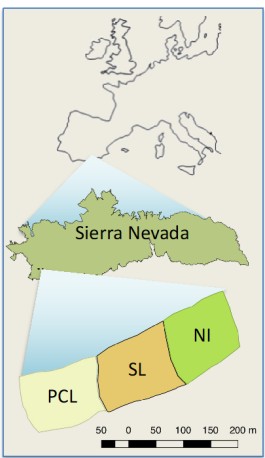

**Figure 1.** Location of the study site within Europe and the Sierra Nevada Protected Area. The three landscape units in which the weeding experiment was conducted were characterized by the post-fire intervention they received: NI = non-intervention unit; PCL = partial cut plus lopping unit; SL = salvage logging unit. The size of each unit was ca. 2 ha (see Table 1 for details).

The burned trees in the NI and PCL landscape units fell during the course of the first post-fire years. The fallen fraction (measured in February each year) was 0.0% in 2006 and 2007, 13.3% in 2008, 84.6% in 2009, and 98.2% in 2010, with similar values in NI and PCL [36]. Accompanying post-fire vegetation was diverse and composed mainly of grasses and forbs, with a mean cover in spring 2007 of 84.0 ± 2.2% for NI, 62.2 ± 3.7% for PCL, and 64.0 ± 5.0% for SL (data for the units used in the present study, extracted from [9]). The most common species were grasses such as *Dactilys glomerata* L. and *Festuca scariosa* (Lag.) Ascherson & Graebner (Poaceae) and forbs such as *Andryala integrifolia* L., *Carlina corymbosa* L., *Scorzonera angustifolia* L. (Asteraceae), *Thapsia villosa* L. (Apiaceae), or *Vaccaria hispanica* (Miller) Rauschert (Caryophyllaceae) (see [9] for detailed list of species).

### 2.3. Effect of Herbaceous Cover on Pine Seedling Survival and Growth

Within each landscape unit, we selected 100 naturally regenerating pine seedlings that were allocated one of two herb-interference treatments: unweeded (no action around pine seedling) or weeded (herbaceous cover removed in a diameter of 50 cm around the pine seedling with a weeding hoe, cutting all the vegetation except the seedling at ca. 5 cm below ground). All pine seedlings had at least 75% herbaceous cover (visual estimation) in a diameter of 1 m around them. The two treatments were intermingled following a stratified design; i.e., one weeded seedling was always followed by an unweeded seedling, and the distance among seedlings was at least 10 m. The seedlings were thus spread across the whole surface of the landscape units. The treatments were implemented in 2007 (second growing season; 2-year-old seedlings hereafter) and again on a different subset of seedlings in 2009 (4-year-old) to test the effect of herb cover at two age stages. Weeding was done in May, at the moment of maximum herbaceous growth in the area. The full experimental design therefore consists of three landscape units, within which each of two herbaceous-cover treatments was applied to 50 seedlings at each of two age classes (total: 600 seedlings; Figure 1).

For each seedling, we measured stem height and stem basal diameter before the onset of the growing season (May 2007 and 2009), and survival, stem height, and stem basal diameter at the end of the corresponding growing season (October). The height of the herbaceous cover around each seedling was measured in May for the pines from the unweeded treatment. Damage by ungulate herbivores during the study years was less than 1% and therefore not further considered [28,37].

### 2.4. Data Analysis

We analyzed the data with linear models in R version 3.3.1 [38] considering seedling survival and growth increment (stem height and diameter) as response variables. In our models, we assessed the effect of weeding (categorical, 2 levels), seedling age (categorical, 2 vs. 4 years), and the weeding by age interaction. We also included the initial size of seedlings as a covariate and all the possible interactions between initial size, weeding, and age to test whether treatment or age effects differed for seedlings of different size. Finally, we included a landscape unit in the models to control the variance generated by this factor—but not to assess effects, as explained below. To assess the significance of effects, we conducted stepwise model simplification [39].

Due to the lack of replication of the three landscape units, we could not analyze the effect of post-fire management regime. Further, the landscape unit is also not suitable as a random factor because three levels are too few to properly estimate the variance produced. Rather, we conducted the weeding treatment across three distinctive landscape units to broaden the population of inference of the weeding experiment, in a way to make results applicable to the three post-fire tree management treatments assessed (independently of possible effects of these treatments). To control for potential effects of landscape units on the data, we included this variable in the models—but, as explained, we do not aim to interpret potential effects as they would be pseudoreplicated [40].

For seedling growth increment, we fitted linear models, and for survival we made a generalized linear model with binomial errors and logit as link function [39]. Additionally, we assessed for correlations between seedling growth and grass height in the unweeded treatment. We estimated Pearson partial correlations between these two variables, accounting for seedling age. As the height of the herbaceous cover surrounding the seedlings was measured after treatment application, it was not possible to use it as a covariate in the overall model. In any case, this would have been inappropriate given the significant positive correlation between herb height and initial seedling size (see Results), as initial seedling size was already included as a covariate in the models.

## 3. Results

### 3.1. Initial Conditions

The height of the herbaceous layer surrounding the seedlings was 14.3 ± 0.6 cm (values are mean ±1 SE throughout the paper) at Year 2 and 22.4 ± 1.0 cm for Year 4. The initial size of the pine seedlings at Year 2 averaged 19.9 ± 0.4 cm for stem height and 5.6 ± 0.1 mm for basal stem diameter, and at Year 4 they averaged 46.2 ± 0.7 cm for height and 12.1 ± 0.2 mm for diameter (see Table 1 for further details). Finally, there was a significant, positive correlation between the maximum height of the herbaceous layer around pine seedlings and initial seedling height each year, with an overall correlation coefficient of 0.36 ($t = 6.67$, $p < 0.001$, $df = 298$; Figure 2). This supports that, at microsite scale, the conditions that were better for the herbaceous vegetation were also better for pine seedlings.

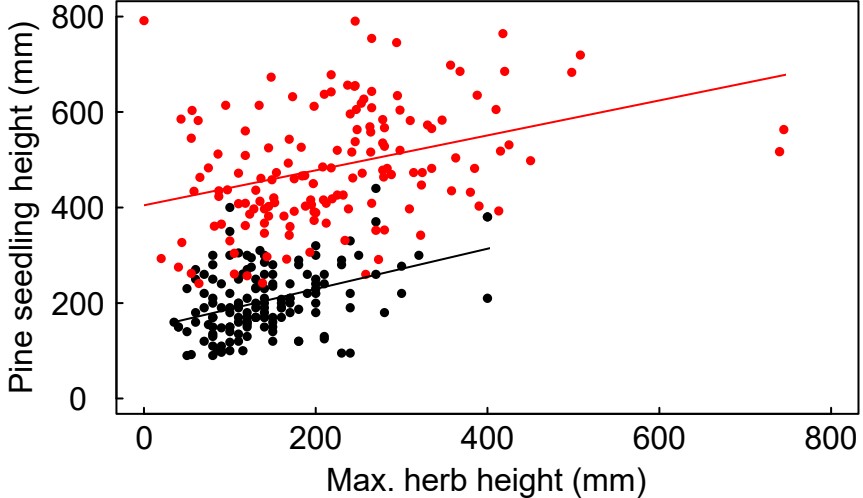

**Figure 2.** Correlation between the height of pine seedlings and the maximum height of the herbaceous layer at the beginning of the sampling season (May, Year 2, and May, Year 4) in a 50 cm diameter around them in the unweeded treatment. The lines show linear regressions for seedlings of age 2 (black) and 4 (red).

### 3.2. Seedling Survival

The weeding treatment improved seedling survival (Table 2). Weeding increased the survival of two-year-old seedlings from an average 44.7% to 67.8%, and this effect was stronger the smaller the seedling (Figure 3a). Four-year-old seedlings registered negligible overall mortality (99.0% survival), so weeding had no effect on these seedlings (Figure 3a). The positive effect of weeding on the survival of two-year-old seedlings was visible at each of the three landscape units, although not significant in the PCL unit if analyzed separately (Figure 3b). Additionally, in two-year-old seedlings, initial seedling size was an important predictor of survival.

**Table 2.** Statistical results.

| Explanatory Variable [a] | | Seedling Survival | | Height Growth | | Stem width Growth | |
|---|---|---|---|---|---|---|---|
| | *df* | $\chi^2$ | *p* | *F* | *p* | *F* | *p* |
| Weeding (W) | 1 | **23.02** | **<0.001** | **4.00** | **<0.05** | [b] | [b] |
| Age (A) | 1 | [b] | [b] | [b] | [b] | **4.87** | **<0.05** |
| Initial size (S) | 1 | [b, c] | [b] | [b] | [b] | [b] | [b] |
| W:A | 1 | 1.80 | 0.18 | 0.01 | 0.92 | 0.23 | 0.63 |
| W:S | 1 | 0.02 | 0.88 | 0.65 | 0.42 | **4.49** | **<0.05** |
| A:S | 1 | **4.88** | **<0.05** | **8.5** | **<0.01** | 1.28 | 0.26 |
| W:A:S | 1 | 0.06 | 0.81 | 0.03 | 0.85 | 0.10 | 0.75 |
| N seedlings | | 596 | | 460 | | 460 | |
| Adjusted $R^2$ [d] | | 0.35 [e] | | 0.10 | | 0.22 | |

[a] Non-significant terms were removed from the model. All models also included the factor post-fire management regime to control for the associated variance; results not shown. [b] Main effects not tested due to inclusion in a significant interaction. [c] The initial size covariate used in these models was seedling height. [d] Estimated for the simplified models, where non-significant terms are excluded. [e] McFadden adjusted pseudo-$R^2$. Significant effects are shown in bold.

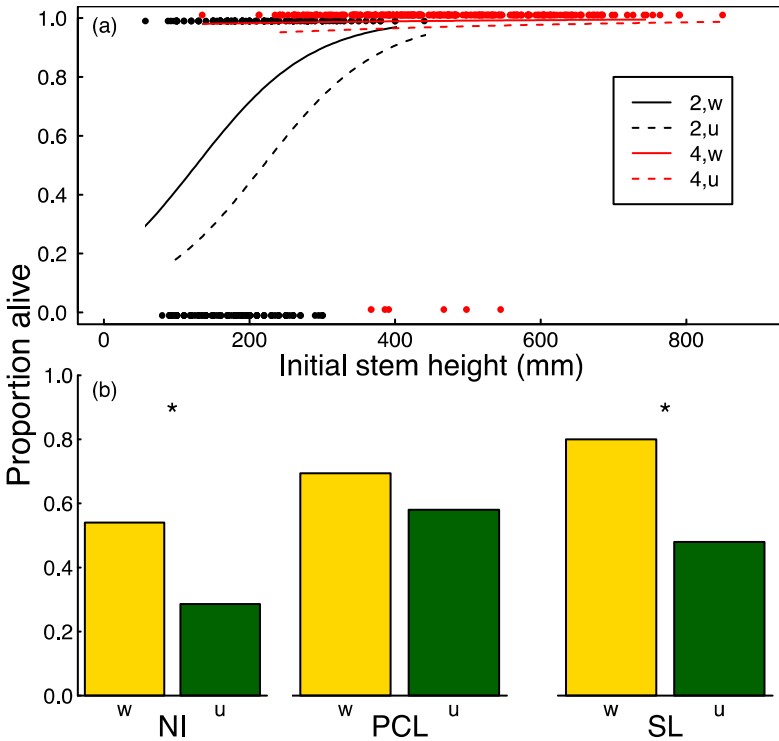

**Figure 3.** Effects of weeding on seedling survival. Panel (**a**) shows the dependence of the weeding effect on initial height and seedling age; lines are based on the parameters from a model that includes all significant terms (Table 2) and pools over landscape units. Similar trends arise when substituting stem height with stem width (data not shown). Panel (**b**) shows the effect of weeding at each landscape unit for two-year-old seedlings (asterisks indicate a significant weeding effect according to a generalized linear model). w = weeded; u = unweeded; 2 and 4 indicate seedling age; NI = non-intervention unit; PCL = partial cut plus lopping unit; SL = salvage logging unit.

*3.3. Seedling Growth*

Weeding increased height growth independently of seedling age or initial size (Table 2), although the magnitude of the increment was small. According to the model, the removal of the herbaceous cover increased stem height increment by an average of 9.4 $\pm$ 4.8 mm (Figure 4a). There was also a positive relationship between initial seedling height and height growth, as taller seedlings grew more

(Figure 4a). Further, the slope of this effect was steeper for two-year-old compared to four-year-old seedlings, which yielded a significant initial height by age interaction (Table 2; Figure 4a).

Growth in stem width was affected by seedling age and by an interaction between weeding and initial stem width (Table 2). Four-year-old seedlings grew more than two-year-old seedlings (Figure 4b). The weeding treatment had a positive effect on the stem width growth of seedlings that already had a thick stem initially, and no effect on seedlings with a small initial stem width (Figure 4b).

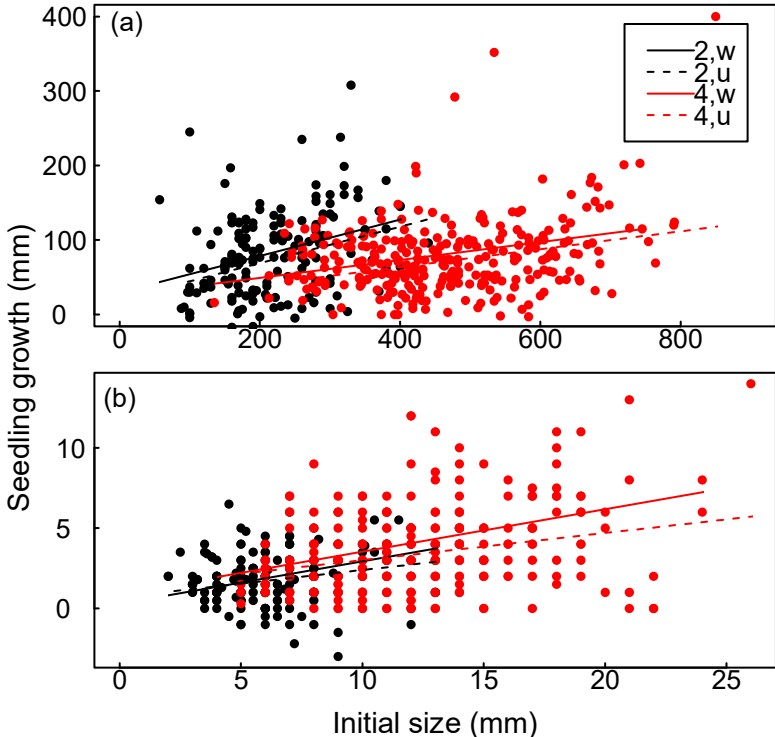

**Figure 4.** Effects of weeding on seedling growth in (**a**) stem height and (**b**) stem width. The lines are drawn from the parameters of a model with all significant terms included, averaged over landscape units. w = weeded; u = unweeded; 2 and 4 indicate seedling age.

## 4. Discussion

Our study shows that the herbaceous layer interfered with the recruitment of a serotinous pine species in a burned Mediterranean ecosystem and that appropriate management can help increase seedling survival and growth. These results may be applicable to many post-fire scenarios, as a dense herbaceous cover usually regenerates quickly after forest fires [31,41]. However, the effect of the herbs changed from negative when pine seedlings were 2 years old to neutral (no effect) two years later, despite the herbs being taller at this later stage. This supports that management actions to promote post-fire forest regeneration should be applied at early stages to be effective or, alternatively, that once a certain threshold is surpassed there is no need for intervention. Our results also suggest that competitive effects may be widespread across different post-fire landscape scenarios, as our study included management extremes from non-intervention to salvage logging that differed in the composition of vegetation [9].

Studies on the effect of the herbaceous layer on tree seedling recruitment may show contrasting results [24,42], and a possible reason underlying such differences is the time at which the interaction is measured [24,43]). Overall, the level of asymmetry created by size-dependent age shifts may be critical for the outcome of plant–plant interactions and resistance to environmental stressors [44,45]). Our results support that this shift in the outcome of the interaction may happen soon in post-fire habitats, as the interference exerted by the herbaceous layer disappeared by the fourth year, and the effect of weeding on seedling growth was relatively small even in the second year. Pulses of nutrient

availability and the lack of a vegetation layer just after fire may explain the high speed with which competitive interactions between ground vegetation and tree seedlings can be overcome. In addition, we found a clear, positive relationship between seedling size at the beginning of the growing season and both the survival and growth of seedlings. This supports that, rather than an effect of age by itself, the key point is to reach a minimum size to surpass the competitive effect of the herbaceous layer.

The competitive effect of the herbaceous layer with tree seedlings is usually ascribed to resource availability for the recruiting seedling, either from the aerial part (higher access to light) or from the subterranean part of the plant (higher access to water and nutrients from the soil; [5,46,47]). In our case, it is not likely that light availability drove the interference between the herbaceous layer and pine seedlings for two reasons. First, the height of the pines was greater than that of the herbs even for two-year-old seedlings (Table 1). Second, light intensity in this Mediterranean-type ecosystem is usually high enough to saturate the photosynthesis [48], and a moderate reduction of radiation by neighbors very often results in facilitative—rather than competitive—interactions in these ecosystems [49,50]. In fact, the reduction in light intensity by the burned wood in the study site resulted in the improvement of the environmental conditions for seedling establishment [28], supporting that light is not a main limiting factor. By contrast, interference at root level has been demonstrated as the main driver of herbs-tree seedling competition even in dense grasslands [51,52]. Larger pine seedlings (both within and across years) likely had a larger root system, which would have reduced competition with the shallow root system of most forbs and grasses [5]. In addition, nutrient acquisition by the pine seedlings may improve with better access to soil water [5,53], thereby reinforcing the positive effect of a larger size.

The positive correlation between the height of the herbaceous layer and that of pine seedlings, which held at both seedling ages, further supports that niche segregation at root level may be a reason underlying the outcome of the interaction in this system. If competition for light were the main driver for herb–pine antagonisms, tall herbs should be expected to reduce light availability most and therefore lead to a smaller tree seedling size. Our result thus likely indicates that differences in the quality of microsites due to other reasons (e.g., water and/or nutrient availability) were larger than the differences in competitive pressure resulting from greater herb biomass. This suggests a strong effect at the microsite level, either due to particular microtopographic environmental features or to legacy effects related to the disturbance (e.g., the presence of coarse woody debris) that promote seedling establishment irrespective of species identity [54,55]. This positive correlation also indicates that the positive effects of initial seedling size on seedling survival and growth may either result from larger seedlings having a greater fitness, or this effect resulting from better microsite conditions, or a combination of both. In either case, positive correlations between initial seedling or sapling size and survival or growth are a common pattern (e.g., [55]), thus supporting the notion that larger seedlings are best suited for post-disturbance tree regeneration, whichever the reason for their greater size.

The results of this study have straightforward practical implications: herbaceous competition may reduce maritime pine performance at the early stages of establishment, but it can be withstood once pine seedlings reach a certain size. As we found effects the second year of recruitment, it is likely that the herbaceous cover will also impact pine seedling recruitment in the first year. In fact, a dense herbaceous cover can even block pine seedling establishment in Mediterranean ecosystems ([56]; see also [31]). On the other side, the age at which the herbaceous layer stops impacting tree seedling performance is likely related to the point at which the interference between root systems is minimized. Thus, although management practices to reduce the impact of the herbaceous layer may depend on a multiplicity of factors, we might establish a guiding threshold given by seedling size and/or time since fire. In our case, it seems plausible that competition is released after four years. Thus, if we were to reduce the impact of competition in regenerating Mediterranean pine forests after fire, management actions (e.g., mowing, herbicide application, etc.) should be done in the first 2–3 years after the fire.

## 5. Conclusions

The herbaceous layer is a source of competition with pine seedlings in Mediterranean post-fire landscapes, and the competition is stronger the smaller/younger the seedling. However, its negative effect may disappear a few years after the fire, probably because of the differentiation of the root systems and consequent below-ground niche partitioning. Management actions to reduce competition from the herbaceous layer should be applied in the second or third year after the fire and to larger seedlings to maximize the efficacy of the treatment. Managing competition by herbs can also be a means to favor the largest pines and thereby increase stand heterogeneity and avoid even-sized regeneration. These results are applicable to a range of post-fire burned-wood management scenarios.

**Author Contributions:** Conceptualization, J.C.; methodology, J.C.; formal analysis A.B.L.; writing—original draft preparation, J.C.; writing—review and editing, J.C. and A.B.L.; project administration, J.C.; funding acquisition, J.C.

**Funding:** This research was funded by projects 10/2005 from the Organismo Autónomo de Parques Nacionales (Spanish Government), CGL2008-01671 from the Spanish Ministerio de Ciencia e Innovación (Spanish Government), and P12-RNM-2705 from Junta de Andalucía (Andalusian regional Government).

**Acknowledgments:** We thank the Consejería de Medio Ambiente, Junta de Andalucía, and the Direction of the Natural and National Park of Sierra Nevada, for fieldwork permission, constant support, and facilities. A.B.L. acknowledges a postdoctoral grant from the Alexander von Humboldt Foundation. We thank two anonymous reviewers for helpful comments that improved the original manuscript.

**Conflicts of Interest:** The authors declare no conflict of interest. The funders had no role in the design of the study; in the collection, analyses, or interpretation of data; in the writing of the manuscript, or in the decision to publish the results.

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
