# Peer review of "Effect of Herbaceous Layer Interference on the Post-Fire Regeneration of a Serotinous Pine (Pinus pinaster Aiton) across Two Seedling Ages"

_forests, doi:10.3390/f10010074_

Reviewer 1 Report

Title – I am not sure “ontogenetic” is the right term to use here. This term is typically used to describe human development. Also, how about mentioning regeneration following fire?   -  Replace “across ontogenetic stages” with “following wildfire”

14 – “gaps in knowledge”

19 – “4 years”  add space

21 – “management of burned forests”

24 – “4-year-old”

24 – replace “who” with “which”

26 – “2-year-old” (here and elsewhere”

32 – Keywords – replace “burnt wood management” with “wildfire management”

Delete “ontogeny”

36 – replace phase with “reforestation and afforestation”. The term  “human-made” is implied by these terms and unnecessary to include.

47 – “also be”

51 – replace “contrasted” with “different”

58 – replace “relax” with “reduce”

59 – replace “through tree ontogeny” with “with increasing tree size”

70 – “burned” is better than “burnt”.  Maybe also considered – “fire-damaged trees”

76 – Replace “fast” with “rapid”

82 – Replace “ontogentic stages” with “different-aged seedlings”

85 and 86 – “2-y-old”    “4-y-old”

91 – Replace “burnt wood” with “post-fire forest”

94-95 – “management scenarios for fire-damaged trees”

99 –Delete “was a” with “included”

Table 1 – include full name with abbreviations of management units in the figure caption.  Also change “herbs” to “herb”

129 – Replace “set up” with ‘design”

186 – Replace “ontogentic stages” with “age classes”

207 – replace “burnt wood” with “fire-damaged tree” (here and elsewhere”

228 and Figure 2 – You now seem to renaming herbaceous vegetation and grass vegetation. Although grass species may be dominant, I would suggest being consistent by using “herbaceous vegetation height”

234 – Be consistent with capitalization of “Year”

Table 2 – Replace “N plants” with “N seedlings”

274 – See earlier suggested wording changes for this section

297 – delete comma

306 – delete “the”

314 – replace “Bigger” with “Larger”  - Also, line 317 and 342

317 – replace “notorious” with “notable”

357 – Not sure if a conclusions section is necessary

402 – “ecosystems”

444 – “and”

Author Response

We appreciate the comments of the reviewer and his/her detailed annotations in the ms. Virtually all changes have been incorporated. The few exceptions where some modification respect to reviewer’s suggestion have been kept are the following:

Line 19: we keep “wood” instead of “forest”, as we refer here to the wood of a single forest.

Line 32. Key-words: we have added “wildfire management” (indicated by the reviewer), but we keep “burnt wood management”. We think this specific term is useful for literature search in this topic. We have removed “ontogeny” according to comments of the two reviewers.

Line 94-95. We think that “fire-damaged trees” is not exactly what we mean, as the trees are already dead in most of the cases. We agree that the previous text may be improved, in any case. We have slightly changed the suggestion of the reviewer: “different management scenarios after fire”.

Line 444. We have not found what the reviewer refers.

Reviewer 2 Report

It is a very interesting paper with valuable results for Forest practice. Even though it is well known that herbaceous plants induce competition on seedlings, the present research adds knowledge to this issue.

In the title, as well as in the manuscript text, I suggest the replacement of the term <> with an age related term. An age difference of two years is not an ontogenic stage difference. Moreover, regarding the discussion section, I think it is rather extended and in some points speculative. In particular, in my opinion, the text in lines 317 – 343 contains speculations in some degree. Based in a positive correlation, authors develop a theoretical  approach that it is not fully supported by the available results. I suggest the reduction of the length of the previous mentioned text.

Finally, in Table I the Pre-treatment mean tree height is 5.8 and 6.1 m in NI and PCL Landscape units respectively. These values are low for mean diameters of 18.9 and 20.1 cm. Please check.

Author Response

We are also grateful to the reviewer for his/her comments, and we have followed all the recommendations:

1. The word “ontogenetic” has been removed from the title and from other parts of the ms, and changed by terms such as age.

2. We have shortened the text from lines 317 to 343 (line numbers referring to original ms), removing some sentences that were too speculative. In addition, we have rephrased some other sentences making them more precise.

3. Table 1. We have double-checked the data of tree height and basal diameter. They are right.